# KnockoffGAN: Generating Knockoffs for Feature Selection using Generative Adversarial Networks

**James Jordon**
Engineering Science Department
University of Oxford, UK
`james.jordon@wolfson.ox.ac.uk`

**Jinsung Yoon**
Department of Electrical and Computer Engineering
UCLA, California, USA
`jsyoon0823@g.ucla.edu`

**Mihaela van der Schaar**
University of Cambridge, UK
Department of Electrical and Computer Engineering, UCLA, California, USA
Alan Turing Institute, London, UK
`mihaela@ee.ucla.edu`

## Abstract

Feature selection is a pervasive problem. The discovery of relevant features can be as important for performing a particular task (such as to avoid overfitting in prediction) as it can be for understanding the underlying processes governing the true label (such as discovering relevant genetic factors for a disease). Machine learning driven feature selection can enable discovery from large, high-dimensional, non-linear observational datasets by creating a subset of features for experts to focus on. In order to use expert time most efficiently, we need a principled methodology capable of controlling the False Discovery Rate. In this work, we build on the promising Knockoff framework by developing a flexible knockoff generation model. We adapt the Generative Adversarial Networks framework to allow us to generate knockoffs with *no assumptions on the feature distribution*. Our model consists of 4 networks, a generator, a discriminator, a stability network and a power network. We demonstrate the capability of our model to perform feature selection, showing that it performs as well as the originally proposed knockoff generation model in the Gaussian setting and that it outperforms the original model in non-Gaussian settings, including on a real-world dataset.

## 1 Introduction

Feature selection is a pervasive problem. Often the goal is to discover features that are relevant to a particular outcome, either for the sake of discovery itself or to aid in prediction [16; 25]. When the focus is on discovery, feature selection methods typically focus on trying to control either the Family-Wise Error Rate (FWER) or the False Discovery Rate (FDR). The FWER measures the probability of making a single false discovery (a Type I error) among the selected features (i.e. selecting one which is not relevant), whereas the FDR measures the proportion of false discoveries made (i.e. the proportion of selected features which are false). Controlling FWER, however, leads to reduced power (i.e. selecting fewer relevant variables) since it controls the probability of making *any* false discovery, whereas FDR tries to control the proportion of false discoveries.

Controlling the FDR is important [5; 6; 3]. Often, data-driven feature selection will be used to select a set of candidate features for further investigation. When further investigation is expensive (for example when further investigation would involve conducting new experiments and collecting more data), a method that cannot control the FDR may result in a large amount of wasted resources, with no guarantee that anything meaningful will be discovered. On the other hand, being able to control the FDR at, say, 10% ensures that at most, 10% of the spent resources are wasted, and 90% are in fact spent on discovering positive, useful results. It should be noted, however, that estimating

the FDR of a method empirically is hard in practice, since we do not have access to the ground truth relevance. As such, a theoretical analysis of the method and its (potential) FDR-controlling properties must be carried out, which does not exist for many existing feature selection methods.

[3] is the seminal paper on the knockoff framework, which is an innovative FDR-controlling feature selection method. Knockoffs are features that are generated to "look like" the real features but be conditionally independent of the label given the real features. Feature statistics (such as the coefficients of a LASSO [32]) are compared between the real features and their knockoffs and a selection is made when this difference is sufficiently large. Performing the selection in this way allows for an estimate of the FDR to be obtained and the selection threshold can be adjusted to control the FDR at the selected level. In the original paper, the relationship between the label and the features is constrained to be of a very specific form; in [7], they remove this constraint and instead provide a theoretical analysis that shifts the burden of knowledge onto knowing the underlying feature distribution. Unfortunately, while the theoretical results hold for any feature distribution, they rely on being able to generate valid knockoffs, for which [7] only provide a method for generating knockoffs when the distribution is a (known) multivariate Gaussian distribution. In this paper, we modify the Generative Adversarial Networks (GAN) [11] framework to address this problem, allowing us to generate knockoffs for any distribution (and without any prior knowledge of it). GANs have been shown to be a powerful method for learning to generate complex distributions [24; 20; 2].

Our main contribution is in modifying the discriminator used in the GAN framework in such a way that the generator learns to generate knockoffs satisfying the necessary swap condition [7] which requires that when a feature and its knockoff are swapped, the joint distribution remains unchanged. In addition, we propose a method for maximizing the power of our model using Mutual Information Neural Estimation (MINE) [4] and investigate a regularization method to improve the stability of training. Our model consists of four networks: (1) a generator network that takes as input noise and the real features, and outputs a set of candidate knockoff features; (2) a discriminator network taking as input "swapped" feature-knockoff features that attempts to determine which variables have been swapped; (3) a Wasserstein GAN discriminator that we use as a regularization term; and (4) a MINE network that estimates the mutual information between each feature-knockoff pair allowing us to maximize the power of the knockoff procedure.

## 2 RELATED WORKS

Feature selection is a well-studied problem with a wealth of related works ([12; 31; 17; 22] provide a summary of a lot of existing literature); however, most methods do not attempt to control the FDR. The most common feature selection method for FDR control is the Benjamini-Hochberg (BHq) procedure and its variants [5; 6], which relies on obtaining valid marginal p-values for each selection.

Knockoffs are an active area of research [9; 18; 10]. The notion of a knockoff was first introduced in [3] with the theory there requiring that the relationship between the features and the label be of a specific form. In [7], they build on the knockoff framework, removing this requirement but instead shifting the requirement to one of knowing the distribution of the features. As noted in the introduction, the theory in [7] holds independent of the distribution of the features - relying only on being able to generate valid knockoffs (which exist for any distribution of features). However, they only propose a method for generating knockoffs when the distribution of features is jointly Gaussian. While they do propose a method for generating approximate knockoffs in the non-Gaussian setting (by simply approximating the features as Gaussian), the guarantees on FDR control do not hold for their approximate knockoffs. In [26] and [10], they add to the class of constructible knockoffs, describing methods for constructing knockoffs for Markov Chains, Hidden Markov Models and Gaussian Mixture Models. Though once again, knowledge of the full distribution is still necessary for their construction.

In this paper we use a framework motivated by GANs [11] to learn to generate knockoffs without *any* assumptions on the distribution of the features. To do this, we modify the discriminator so that rather than trying to determine whether a sample is real or fake, it attempts to identify which components have been "swapped". In [38], an unconventional discriminator is used that performs component-wise discrimination for the purpose of imputation. While the problem addressed in that paper is different to the one here, the key idea relies on a similar modification to the discriminator to be able to appropriately guide the generator.

In order to maximize the power of our variable selection mechanism, it will be desirable that the feature-knockoff pairs are "as independent as possible" (this is discussed in [7]). In order to achieve this we will investigate the use of a promising recent paper, MINE [4]. MINE proposes a neural architecture and training procedure capable of estimating the mutual information between two random variables. As the mutual information between two random variables is zero only when they are independent, we will use this as a measure of independence and attempt to minimize it during the training of our modified GAN.

## 3 BACKGROUND

In this section we introduce our notation and define knockoffs as in [7]. Let us denote the feature space by $\mathcal{X}$ and the label space by $\mathcal{Y}$. Let the dimension of $\mathcal{X}$ be $d$. Suppose that $\mathbf{X} = (X_1, ..., X_d)$ and $Y$ are random variables over $\mathcal{X}$ and $\mathcal{Y}$. As in [7], we will work with the notion of a *null set*.

**Definition 1.** *A variable $X_j$ is said to be "null" if and only if $Y$ is independent of $X_j$ conditional on $\{X_i : i \neq j\}$. We define $\mathcal{H}_0$ to be the set of all null variables.*

Our goal will be to discover as many relevant features as possible while controlling the FDR. For a given (potentially random) selection procedure that selects $\hat{\mathcal{S}} \subset \{1, ..., d\}$, we define the FDR to be

$$\text{FDR} = \mathbb{E}\left[\frac{|\hat{\mathcal{S}} \cap \mathcal{H}_0|}{|\hat{\mathcal{S}}|}\right].$$

Note that this agrees with the usual notion of FDR (i.e. when defined in terms of the Markov blanket) under mild assumptions (for a more thorough discussion see [7]).

### 3.1 KNOCKOFFS

**Definition 2.** *A knockoff [7] for $\mathbf{X}$ is a random variable $\tilde{\mathbf{X}} \in \mathcal{X}$ satisfying the following two properties:*

$$(\mathbf{X}, \tilde{\mathbf{X}}) \overset{d.}{=} (\mathbf{X}, \tilde{\mathbf{X}})_{swap(S)} \tag{1}$$

$$\tilde{\mathbf{X}} \perp\!\!\!\perp Y | \mathbf{X} \tag{2}$$

*for all $S \subset \{1, ..., d\}$ where $(\cdot, \cdot)_{swap(S)}$ denotes the vector obtained by swapping the $i$th component with the $(i + d)$th component for each $i \in S$ and $\overset{d.}{=}$ is equality in distribution.*

In order to use knockoffs for feature selection, we must define an appropriate feature statistic, $W_j$, that depends on $\mathbf{X}, \tilde{\mathbf{X}}$ and $Y$, i.e. $W_j = w_j((\mathbf{X}, \tilde{\mathbf{X}}), Y)$ for some function $w_j$. This function $w_j$ must satisfy the following flip-sign property:

$$w_j((\mathbf{X}, \tilde{\mathbf{X}})_{\text{swap}(S)}, Y) = \begin{cases} w_j((\mathbf{X}, \tilde{\mathbf{X}}), Y) \text{ if } j \notin S \\ -w_j((\mathbf{X}, \tilde{\mathbf{X}}), Y) \text{ if } j \in S. \end{cases} \tag{3}$$

One of the procedures used in [7] to construct these statistics is to perform LASSO, treating the augmented feature-knockoffs as the features on which to regress. This gives LASSO coefficients $b_1, ..., b_{2d}$, and the statistic $W_j$ is set to be the LASSO Coefficient Difference given by

$$W_j = |b_j| - |b_{j+d}|.$$

Note that the FDR control guarantees hold independently of the choice of statistic, but a poorly chosen statistic can significantly impact the power of the test. In particular, using the LASSO Coefficient Difference in non-linear settings can yield few discoveries. The focus of this paper, however, is on generating the knockoffs, not on the statistic used on top of the generated knockoffs and so in our synthetic experiments, we use a linear model for $Y$ to be able to draw fair comparisons between our model and [7]. In the real world data experiment, we use a statistic based on Random Forests for both methods [37].

The following result from [7] depends only on having obtained knockoffs that satisfy Definition 2 and feature statistics satisfying (3) (and in particular do not depend specifically on using LASSO to obtain the statistics).

**Theorem 1.** *Let $q \in [0, 1]$. Given test statistics, $W_1, ..., W_d$, satisfying (3), let*

$$\tau = \min \left\{ t > 0 : \frac{1 + |\{j : W_j \leq -t\}|}{|\{j : W_j \geq t\}|} \leq q \right\}.$$

*Then the procedure selecting the variables*

$$\hat{\mathcal{S}} = \{j : W_j \geq \tau\}$$

*controls the FDR at level $q$, i.e.*

$$\mathbb{E}\left[ \frac{|\hat{\mathcal{S}} \cap \mathcal{H}_0|}{|\hat{\mathcal{S}}| \vee 1} \right] \leq q.$$

## 4 KNOCKOFFGAN

It should be noted that in order to satisfy equation (2), it simply needs to be the case that the knock-offs are constructed without looking at the label, $Y$. In order to satisfy equation (1) we use a modified GAN framework, which gives us the flexibility to learn to generate knockoffs without any assumptions on the distribution of the original features.

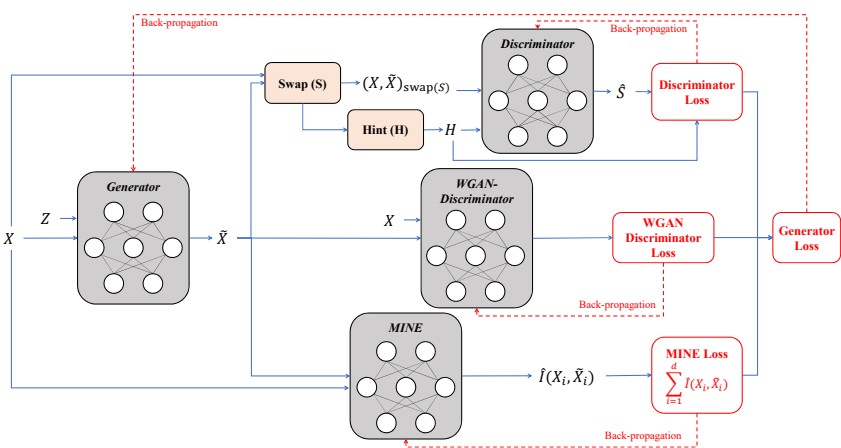

Figure 1: KnockoffGAN Block Diagram

### 4.1 GENERATOR

The generator, $G$, will be a function $G(\cdot, \cdot; \phi) : \mathcal{X} \times [0, 1]^c \to \mathcal{X}$, parametrized by $\phi$ that takes a realization $\mathbf{x}$ of $\mathbf{X}$ and random noise, $\mathbf{z} \sim \mathcal{U}([0, 1]^c)$, as inputs and outputs knockoff features $\tilde{\mathbf{x}}$. We define $\tilde{\mathbf{X}} := G(\mathbf{X}, \mathbf{z})$. We model $G$ as a fully connected neural network with weights $\phi$.

### 4.2 DISCRIMINATOR

The main innovation of our paper is in defining the discriminator. Equation (1) imposes a condition on the joint distribution of $(\mathbf{X}, \tilde{\mathbf{X}})$ and as such we must define a discriminator with a loss that is (not necessarily uniquely) minimized only for joint distributions satisfying this condition. To that end, the discriminator, $D$, will be a function $D(\cdot; \psi) : \mathcal{X} \times \mathcal{X} \to [0, 1]^d$ that takes as input a *swapped* sample-knockoff pair $(\mathbf{x}, \tilde{\mathbf{x}})_{\text{swap}(S)}$ and outputs a vector in $[0, 1]^d$ with the $i$th component of $D((\mathbf{x}, \tilde{\mathbf{x}})_{\text{swap}(S)})$ corresponding to the probability that $i \in S$. The discriminator is attempting to detect which variables have been swapped and, intuitively, when the discriminator is unable to determine this, the swapped and unswapped joint distributions must be the same.

The loss we use to train the discriminator is the multi-output cross-entropy loss given by

$$\mathcal{L}_D = \sum_{\mathbf{S} \in \{0,1\}^d} \mathbb{E}_{\mathbf{X} \sim \mathcal{P}_\mathbf{X}}[\mathbb{E}_{\tilde{\mathbf{X}} \sim \tilde{\mathcal{P}}_\mathbf{X}(\mathbf{X})}[\mathbf{S} \cdot \log(D((\mathbf{X}, \tilde{\mathbf{X}})_{\text{swap}(S)})) + (\mathbf{1} - \mathbf{S}) \cdot \log(1 - D((\mathbf{X}, \tilde{\mathbf{X}})_{\text{swap}(S)}))]]$$

$$(4)$$

where $\cdot$ is the standard dot, $\mathbf{1} = (1, ..., 1)$, $\mathbf{S} = (S_1, ..., S_d)$ with $S_i = \mathbb{I}(i \in S)$ ($\mathbb{I}$ is the indicator function) and $\log$ is taken element-wise. The following theorem is our main theoretical result, which states that the training regime employed by KnockoffGAN will result in a procedure that generates valid knockoffs.

**Theorem 2.** *Equation (4) is maximized (with respect to G) if and only if equation (1) is satisfied by G.*

*Proof.* The proof, alongside supporting theoretical results, can be found in the Appendix. $\quad\square$

In practice, the sum is too computationally expensive ($O(2^d)$) to calculate and so we perform stochastic gradient descent using minibatches with $\mathbf{S}$ sampled uniformly from $\{0, 1\}^d$, independently for each sample in the minibatch.

We also found that training with respect to the full loss resulted in a poor performance, particularly when $d$ is large. We found that the discriminator struggled to learn anything when asked to find the full swap vector, and the poor discriminator resulted in a poorly trained generator. In order to overcome this, we introduce a hint vector - first introduced in [38] - that we use to reveal partial information to the discriminator about the swap vector. We do this by using the hint to reveal some, but not all, of the components of $S$ to the discriminator. In doing so, we reduce the burden of the discriminator from needing to determine the entire swap vector to only needing to determine *some* of the swap vector.

Formally, the hint, $\mathbf{H}$, will be a random variable depending on $\mathbf{S}$, that we pass to the discriminator, alongside $(\mathbf{X}, \tilde{\mathbf{X}})_{\text{swap}(S)}$. We use the hint to control the amount of information we pass to $D$ about $\mathbf{S}$ before asking $D$ to predict $\mathbf{S}$. In practice, our hinting mechanism involves sampling a multivariate Bernoulli random variable, $\mathbf{B}$ from i.i.d. components, which each take value $1$ with probability $0.9$. The hint is then constructed by setting $H_i = S_i$ if $B_i = 1$ and $H_i = 0.5$ if $B_i = 0$. The discriminator is therefore being asked only to predict the values of $\mathbf{S}$ for which $B_i = 0$; the others, $D$ is able to directly infer from $H_i$. In order to avoid overfitting to the hint, it becomes necessary to remove these terms from our loss. Our loss now becomes

$$\mathcal{L}_D = \sum_{\mathbf{S} \in \{0,1\}^d} \mathbb{E}_{\mathbf{X} \sim \mathcal{P}_\mathbf{X}} \big[ \mathbb{E}_{\tilde{\mathbf{X}} \sim \tilde{\mathcal{P}_\mathbf{X}}(\mathbf{X})} \big[ \mathbb{E}_{\mathbf{H} \sim \mathcal{P}_{\mathbf{H}|\mathbf{S}}} \big[ (\mathbf{S} \odot (\mathbf{1} - \mathbf{B})) \cdot \log(D((\mathbf{X}, \tilde{\mathbf{X}})_{\text{swap}(S)}, \mathbf{H})) \quad (5)$$

$$+ ((\mathbf{1} - \mathbf{S}) \odot (\mathbf{1} - \mathbf{B})) \cdot \log(1 - D((\mathbf{X}, \tilde{\mathbf{X}})_{\text{swap}(S)}, \mathbf{H})) \big] \big] \big]$$

where $\odot$ denotes element-wise multiplication and the expectation over $\mathbf{B}$ is implicit in the expectation over $\mathbf{H}$.

### 4.3 STABILITY

We found that adding a regularization term in the form of a Wasserstein GAN discriminator (with Gaussian Process (GP) regularization) [2], $f$, aided performance. We note that when equation (1) holds, we must have that $\mathbf{X} \stackrel{d.}{=} \tilde{\mathbf{X}}$ and so the addition of this regularizing term does not affect the optimal solution to our loss. We model $f$ as a fully connect neural network with weights $\nu$. The loss is given by

$$\mathcal{L}_f = \mathbb{E}\left[ f(\mathbf{X}) - f(\tilde{\mathbf{X}}) - \eta(||\nabla_{\hat{\mathbf{X}}} f(\hat{\mathbf{X}})||_2 - 1)^2 \right]$$

where $\epsilon \sim \mathcal{U}[0, 1]$, $\hat{\mathbf{X}} = \epsilon \mathbf{X} + (1 - \epsilon)\tilde{\mathbf{X}}$ and $\eta$ is a hyper-parameter (set to 10 in practice). Note that we have rewritten the loss to be the negative of the one given in [2], allowing us to write our overall objective as a minimax problem. This loss is added to the generator loss as an additional regularization term.

### 4.4 MAXIMIZING POWER

As noted in [7], it is intuitive that in order to maximize the power of the knockoff selection procedure, we wish to make $X_j$ and $\tilde{X}_j$ as "independent" as possible. Doing so ensures that as little as possible of the dependence between the real feature and the label is present between the knockoff and the label; this allows us to determine whether or not the relationship between the feature and label is only through the feature's correlation with other features, or is in fact a true signal.

In order to achieve maximal independence, we look to minimize the mutual information between each feature and its knockoff. Actually computing the true mutual information requires access to both the joint density of the feature-knockoff pairs and to the marginal densities of each feature and knockoff, which we do not have.

Instead, we look to a promising recent work, Mutual Information Neural Estimation (MINE [4]), that provides a framework for estimating the mutual information using neural networks. To do so, they estimate the mutual information between random variables $U$ and $V$ by performing gradient ascent on the following objective:

$$\sup_{\theta \in \Theta} \mathbb{E}_{\mathbb{P}_{UV}^{(n)}}[T_\theta] - \log(\mathbb{E}_{\mathbb{P}_U^{(n)} \otimes \mathbb{P}_V^{(n)}}[e^{T_\theta}])$$

where $\mathbb{P}_{UV}$ denotes the joint measure of $(U, V)$ with $\mathbb{P}_U = \int_{\mathcal{V}} d\mathbb{P}_{UV}$ and $\mathbb{P}_V = \int_{\mathcal{U}} d\mathbb{P}_{UV}$ denoting the marginal measures. $^{(n)}$ denotes the empircal distribution associated with $n$ i.i.d. samples.

Using MINE we approximate the mutual information between each pair $X_j$ and $\tilde{X}_j$ by using $d$ neural networks[1], $T^1, ..., T^d$, each parametrized by $\theta_1, ..., \theta_d$, that we refer to collectively as the *power network*, and will write $P$ to denote the collection of networks $T^1, ..., T^d$. The mutual information is added using a trade-off parameter $\lambda$ to the loss for $G$. Formally, define $\mathcal{L}_P$ by

$$\mathcal{L}_P = \sum_{j=1}^{d} \left( \sum_{i=1}^{n} (T_{\theta_j}^j(x_j^{(i)}, \tilde{x}_j^{(i)})) - \log(\sum_{i=1}^{n} \exp(T_{\theta_j}^j(x_j^{(\kappa(i))}, \tilde{x}_j^{(i)}))) \right)$$

where $\kappa$ is a random permutation of $[n]^2$ and $^{(i)}$ denotes the $i$th sample - noting that dependence on $G$ is through $\tilde{\mathbf{X}}$.

### 4.5 FINAL OBJECTIVE

The resulting minimax game played by $G$, $D$, $W$ and $P$ is given by

$$\min_G \left( \max_D(\mathcal{L}_D) + \lambda \max_P(\mathcal{L}_P) + \mu \max_f(\mathcal{L}_f) \right)$$

where $\lambda, \mu$ are hyper-parameters (set to 1 in the experiments section).

We train each of $G$, $D$, $W$ and $P$ iteratively. Pseudo-code of our knockoff construction algorithm can be found in Algorithm 1 and a visual representation of our architecture in Fig. 1.

After generating knockoffs, feature statistics are computed according to some procedure (in the synthetic experiments we use LASSO and in the real data experiment we use a Random Forest-based statistic [37]). Features are then selected based on these statistics according to Theorem 1.

## 5 EXPERIMENTS

In this section we demonstrate the capability of our method to match the results of [7] in settings where their model is correctly specified (i.e. when the underlying feature distribution is Gaussian) and then go on to show, in settings where the underlying feature distribution is non-Gaussian, that our method is able to outperform their Gaussian approximation. We compare to two versions of the BHq method [5; 6] to provide a baseline.

We also perform a qualitative analysis of KnockoffGAN on a real-world dataset. We compare features found by KnockoffGAN to PubMed literature and show that KnockoffGAN discovers several meaningful features for 2 different disease outcomes.

---

[1]In practice we use a single neural network with diagonalized weights to parallelize these networks.

[2]In our pseudo-code, we write $\mathcal{U}(S_n)$ to denote the uniform distribution over the set of all permutations of $[n] = \{1, ..., n\}$.

---

**Algorithm 1** Pseudo-code of KnockoffGAN

1: **Inputs:** mini-batch size $n_{mb} > 0$, **Initialize** parameters $\phi, \psi, \nu, \theta_1, ..., \theta_d$
2: **while** Converge **do**
3:     **Discriminator Update**
4:         Sample $\mathbf{x}_1, ..., \mathbf{x}_{n_{mb}}$ from $\mathcal{D}$, $\mathbf{z}_1, ..., \mathbf{z}_{n_{mb}} \sim \mathbb{P}_Z$
5:         Sample $\mathbf{S}_1, ..., \mathbf{S}_{n_{mb}} \overset{i.i.d}{\sim} \mathcal{U}(\{0,1\}^d)$, $\mathbf{b}_1, ..., \mathbf{b}_{n_{mb}} \sim \text{Ber}(0.9)$
6:         **for** $i = 1, ..., n_{mb}$ **do**
7:             $\tilde{\mathbf{x}}_i \leftarrow G(\mathbf{x}_i, \mathbf{z}_i; \phi)$
8:             $\mathbf{h}_i = \mathbf{S}_i \odot \mathbf{b}_i + 0.5(\mathbf{1} - \mathbf{b}_i)$
9:         Update $D$ by ascending its stochastic gradient

$$\nabla_\psi \sum_{i=1}^{n_{mb}} \Big[ (\mathbf{S}_i \odot (\mathbf{1} - \mathbf{b}_i)) \cdot \log(D((\mathbf{x}_i, \tilde{\mathbf{x}}_i)_{\text{swap}(\mathbf{S})}), \mathbf{h}_i)$$
$$+ ((\mathbf{1} - \mathbf{S}_i) \odot (\mathbf{1} - \mathbf{b}_i)) \cdot \log(\mathbf{1} - D((\mathbf{x}_i, \tilde{\mathbf{x}}_i)_{\text{swap}(\mathbf{S})}), \mathbf{h}_i)) \Big]$$

10:     **MINE Update**
11:         Sample $\mathbf{x}_1, ..., \mathbf{x}_{n_{mb}}$ from $\mathcal{D}$, $\mathbf{z}_1, ..., \mathbf{z}_{n_{mb}} \sim \mathbb{P}_Z$, $\kappa \sim \mathcal{U}(S_{n_{mb}})$
12:         **for** $i = 1, ..., n_{mb}$ **do**
13:             $\tilde{\mathbf{x}}_i \leftarrow G(\mathbf{x}_i, \mathbf{z}_i; \phi)$
14:         **for** $j = 1, ..., d$ **do**
15:             Update $T_j$ by ascending its stochastic gradient
$$\nabla_{\theta_j} \Big( \sum_{i=1}^{n_{mb}} T_{\theta_j}^j(x_j^{(i)}, \tilde{x}_j^{(i)}) \Big) - \log \Big( \sum_{i=1}^{n_{mb}} \exp(T_{\theta_j}^j(x_j^{(i)}, \tilde{x}_j^{(\kappa(i))})) \Big)$$

16:     **WGAN-GP Update**
17:         Sample $\mathbf{x}_1, ..., \mathbf{x}_{n_{mb}}$ from $\mathcal{D}$, $\mathbf{z}_1, ..., \mathbf{z}_{n_{mb}} \sim \mathbb{P}_Z$
18:         **for** $i = 1, ..., n_{mb}$ **do**
19:             Sample $\epsilon \sim \mathcal{U}[0,1]$
20:             $\tilde{\mathbf{x}}_i \leftarrow G(\mathbf{x}_i, \mathbf{z}_i; \phi)$
21:             $\hat{\mathbf{x}}_i = \epsilon \mathbf{x}_i + (1 - \epsilon) \tilde{\mathbf{x}}_i$
22:         Update $f$ by ascending its stochastic gradient
$$\nabla_\nu \sum_{i=1}^{n_{mb}} \Big[ f(\mathbf{x}_i) - f(\tilde{\mathbf{x}}_i) - \eta(||\nabla_{\hat{\mathbf{x}}_i} f(\hat{\mathbf{x}}_i)||_2 - 1)^2 \Big]$$

23:     **Generator Update**
24:         Sample $\mathbf{x}_1, ..., \mathbf{x}_{n_{mb}}$ from $\mathcal{D}$, $\mathbf{z}_1, ..., \mathbf{z}_{n_{mb}} \sim \mathbb{P}_Z$
25:         Sample $\mathbf{S}_1, ..., \mathbf{S}_{n_{mb}} \overset{i.i.d}{\sim} \mathcal{U}(\{0,1\}^d)$, $\kappa \sim \mathcal{U}(S_{n_{mb}})$
26:         **for** $i = 1, ..., n_{mb}$ **do**
27:             $\tilde{\mathbf{x}}_i \leftarrow G(\mathbf{x}_i, \mathbf{z}_i; \phi)$
28:         Update $G$ by descending its stochastic gradient
$$\nabla_\phi(\mathcal{L}_D + \lambda \mathcal{L}_P + \mu \mathcal{L}_f)$$

---

## 5.1 SYNTHETIC DATA EXPERIMENTS

### 5.1.1 SIMULATION SETTINGS

Evaluating feature selection methods on real data is difficult as we do not have access to the ground truth. To evaluate KnockoffGAN, we conduct a series of experiments using *synthetic data*, replicating those carried out in [7] and extending them to more general settings. In each of the following synthetic experiments, we set the feature dimension to be $d = 1000$ and the number of samples to be $n = 3000$. For each feature distribution we perform two experiments:

1. Y-Logit: $P(Y = 1 | \mathbf{X}) = \frac{\exp(m(\mathbf{X}))}{(1 + \exp(m(\mathbf{X})))}$

2. Y-Gaussian: $Y \sim \mathcal{N}(m(\mathbf{X}), 1)$

where $m(\mathbf{X}) = \sum_{i=1}^{60} \alpha \delta_i X_i$ with $\delta_i \in \{-1, 1\}$ sampled uniformly and then fixed for each experiment. $\alpha$ controls the strength of the influence that $\mathbf{X}$ has on $Y$, and in the experiments we vary this (as in [7]). Note that for the auto-regressive settings (found in Section 5.1.2 and the Appendix) the relevant variables are sampled uniformly at random from among the 1000 features (rather than being the first 60); in the non-auto-regressive settings this is not necessary.

We report the True Positive Rate (TPR), which is also commonly referred to as the *power*, defined as

$$\text{TPR} = \frac{|\hat{\mathcal{S}} \cap \mathcal{S}^*|}{|\mathcal{S}^*|} \tag{6}$$

where $\mathcal{S}^* = \{1, ..., d\} \setminus \mathcal{H}_0$ is the set of all non-null features. We also report the FDR to verify that the methods do indeed control it at the specified level which we set to be 10%. Note that we are not using FDR as a metric - a lower FDR is not desirable when we set the threshold to 10%. In fact, we want the methods to be as close to 10% as possible (so that they are achieving maximum power). We perform 100 replications of each experiment and report the average TPR and FDR.

### 5.1.2 GAUSSIAN SETTINGS

We begin by replicating the setup from [7] in which the underlying feature distribution is Gaussian. In this setting, we do not expect KnockoffGAN to perform better than the original knockoff framework as the original framework assumes a Gaussian distribution. Our goal here is simply to achieve a similar performance, demonstrating that little performance is lost even when the distribution is known to be Gaussian.

In the first experiment that we replicate from [7], the features are set to be auto-regressive (AR(1)) with Gaussian marginal distributions, i.e. $X_i = \phi X_{i-1} + Z_i$ with $Z_i$ being chosen such that $X_i \overset{i.i.d.}{\sim} \mathcal{N}(0, \frac{1}{n})$. In this experiment we vary $\phi$, which determines the correlation between features, rather than $\alpha$. We fix $\alpha = 3.5$ for Y-Gaussian and $\alpha = 10$ for Y-Logit. The results are reported in Fig. 2.

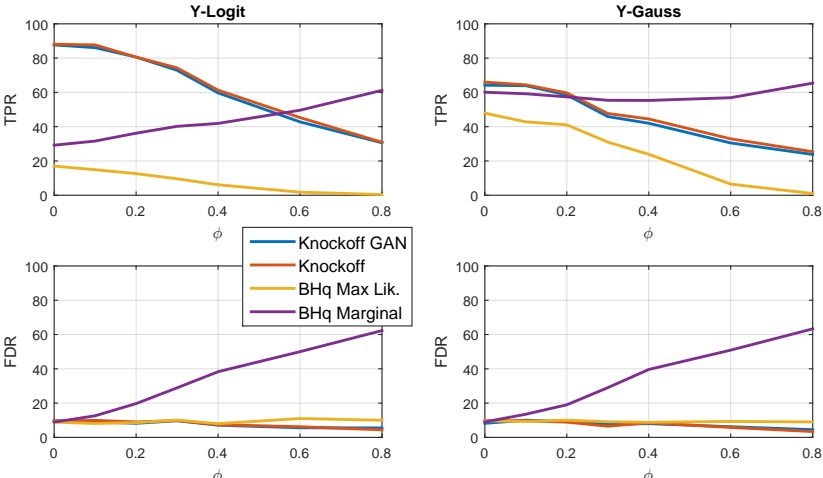

Figure 2: Comparison of KnockoffGAN with the benchmarks for $\mathbf{X}$ distributed as an auto-regressive distribution with Gaussian marginal distributions. TPR is used to quantify performance and FDR is reported to verify that it is at the specified threshold (10%).

As in [7], we observe that BHq Marginal, which tests for marginal independence of the feature from $Y$, suffers from severely increased FDR as we increase the correlation, invalidating the seemingly good TPR. To make the remaining results clearer, we omit BHq marginal from the rest of this section. Aside from this, we see in Fig. 2, that the other methods control the FDR at or very close to the specified 10% threshold. We also see that across the entire range of $\alpha$, KnockoffGAN achieves a very similar TPR to the original Knockoff framework.

In the second experiment, we set the underlying feature distribution to be i.i.d. Gaussian. We found in this case also that KnockoffGAN was able to control the FDR and achieve a similar TPR to the

original knockoff framework. More details of this experiment and the results for it can be found in the Appendix

### 5.1.3 Non-Gaussian settings

We now move on to the key results for the paper in which the underlying feature distribution is no longer Gaussian. In this setting, we expect to outperform the original Knockoff framework due to the fact that they approximate the distribution as Gaussian. In particular, when this approximation is poor, the knockoffs are no longer valid and as such no FDR guarantees can be given. On the other hand, KnockoffGAN does not place any requirements on the distribution of the features and as such is able to generate valid knockoffs.

We performed experiments for several different underlying feature distributions, and found that KnockoffGAN achieved a higher TPR than the original knockoff framework in all cases, while controlling the FDR at the specified level. We give the results for $\mathbf{X}$ coming from a 4-Gaussian mixture model in Fig. 3 - results for Uniform, Dirichlet, and other (2 and 3) Gaussian mixture models can be found in the Appendix.

To create our 4-mixture model, we set the means $(\mathbf{m}^1, \mathbf{m}^2, \mathbf{m}^3, \mathbf{m}^4)$ of the 4 Gaussians to be:

- $m_i^1 = 1$ for $i = 1$ to 100 and 0 for $i = 101$ to 1000,
- $m_i^2 = 1$ for $i = 1$ to 50 and -1 for $i = 51$ to 100 and 0 for $i = 101$ to 1000,
- $m_i^3 = -1$ for $i = 1$ to 50 and 1 for $i = 51$ to 100 and 0 for $i = 101$ to 1000,
- $m_i^4 = -1$ for $i = 1$ to 100 and 0 for $i = 101$ to 1000.

We scale the variance of each Gaussian to be such that the overall variance of each feature is $\frac{1}{n}$.

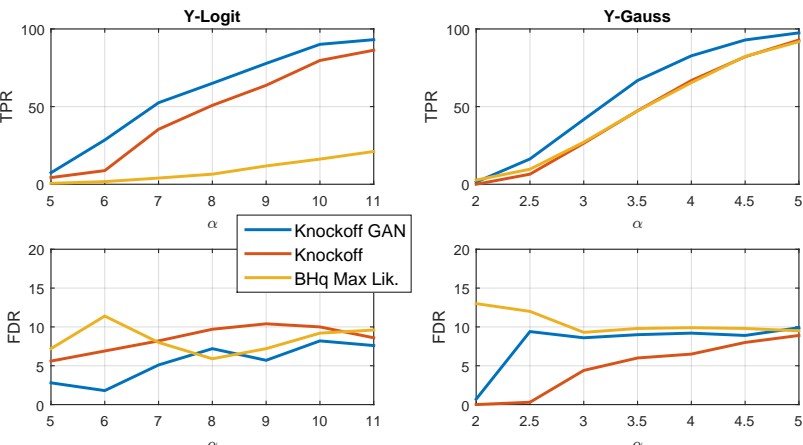

Figure 3: Comparison of KnockoffGAN with the benchmarks for $\mathbf{X}$ distributed as a 4-mixture Gaussian mixture model. TPR is used to quantify performance and FDR is reported to verify that it is at the specified threshold (10%).

We see in Fig. 3 that KnockoffGAN consistently outperforms the original knockoff framework, achieving a higher TPR across the entire range of $\alpha$ while consistently controlling the FDR at 10%. In fact, in the $Y$-Gaussian setting we see that the original knockoff framework performs almost identically to BHq Maximum Likelihood.

### 5.1.4 Impact of WGAN regularization

We conclude the synthetic experiments by demonstrating the effect of the WGAN regularizer[3]. We conduct this experiment using an auto-regressive model with $\mathcal{U}(-\sqrt{3/n}, \sqrt{3/n})$ marginal distributions. We fix $\alpha = 5$ for Y-Logit and $\alpha = 2.5$ for Y-Gauss.

---

[3]The WGAN regularizer was included in all previous experiments.

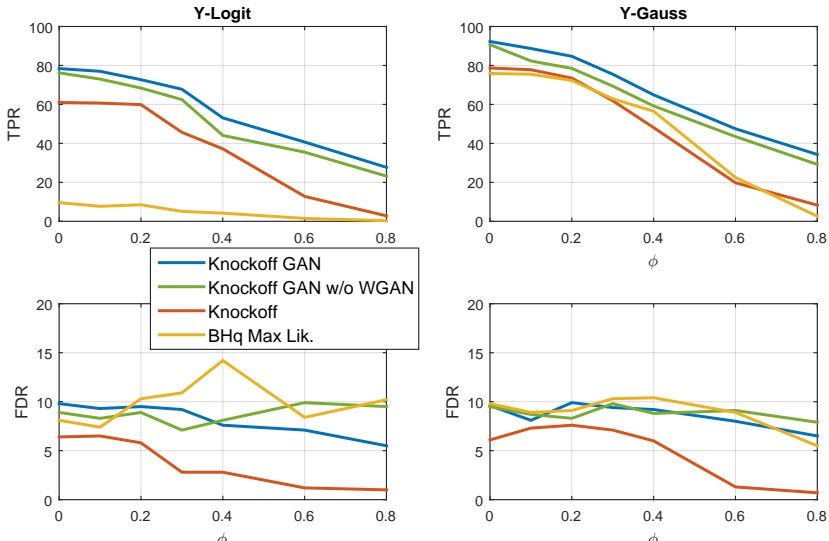

Figure 4: A comparison of the performance of KnockoffGAN with and without the WGAN regularizer for $\mathbf{X}$ distributed as an auto-regressive distribution with $\mathcal{U}(-\sqrt{3/n}, \sqrt{3/n})$ marginal distributions. TPR is used to quantify performance and FDR is reported to verify that it is at the specified threshold (10%).

As we see in Fig. 4, the WGAN regularizer has a significant effect on the results, with the improvement in some places being almost as much as KnockoffGAN without WGAN makes over the original knockoff framework. As noted in Section 4.3, there is no trade-off introduced by the inclusion of this regularizer; the optimal solution to the loss is unchanged and therefore this regularization is "free" in terms of FDR control, but as demonstrated improves TPR performance.

## 5.2 REAL DATA EXPERIMENT

In this section we use a biobank dataset (with 387 dimensions) to qualitatively analyze the performance of KnockoffGAN. We use KnockoffGAN to select features for two different outcomes: (1) Cardiovascular Disease (CVD) and (2) Diabetes and then use PubMed literature to asses the validity of the selected features.

We found that the original knockoff framework was unable to select even the most well-known features (such as Age and Sex for CVD [15]), even when the FDR threshold was increased to 20%. Therefore, there are no relevant features to report for the original knockoff framework and so Table 1 contains only the features selected by KnockoffGAN that were deemed relevant by PubMed literature. For this the FDR threshold was set to 5% so that the number of discoveries was manageable for cross-reference with PubMed.

| No | Cardiovascular Disease | PubMed ref. | Diabetes | PubMed ref. |
|----|------------------------|-------------|----------|-------------|
| 1 | Age | [15] | Lipid-lowering drugs | [35] |
| 2 | Sex | [15] | Comparative body size | [27] |
| 3 | Daily smoking | [1] | Home owned | [13] |
| 4 | FEV1 | [28] | Insomnia | [34] |
| 5 | Diastolic blood pressure | [33] | Anti-hypertensive drugs | [8] |
| 6 | Diabetes | [29] | Asthma | [30] |
| 7 | Father chronic bronchitis | [14; 23] | Height | [19; 27] |
| 8 | Alcohol intake | [21] | Alcohol intake | [36] |
| 9 | Long-standing illness | | | |

Table 1: Discovered features using KnockoffGAN framework, verified using PubMed literature. The FDR threshold was set to 5%.

As we see in Table 1, KnockoffGAN discovers 9 relevant features for CVD and 8 relevant features for diabetes. Some of the relevant features, such as Age, Sex and Long-standing illness for CVD are

well-known relevant features. The remaining features are all supported by the literature in PubMed. While this is a qualitative result (it relies on using PubMed as the ground truth), we do believe this demonstrates that KnockoffGAN is a significant improvement over the original knockoff generation procedure.

# 6    CONCLUSION

In this paper we built on the knockoff framework introduced in [3] by developing a novel GAN framework, KnockoffGAN, capable of generating knockoffs with no assumptions on the underlying data. We demonstrated through a series of experiments on a range of synthetic datasets and on a real world dataset that our method improves on the performance of the original knockoff framework.

While we feel this is a significant step towards being able to generate knockoffs for any data, there is still more work to be done. In particular, generalizing this method to time-series data would be non-trivial, and would be an interesting avenue for further investigation.

## ACKNOWLEDGEMENT

The authors would like to thank the reviewers for their helpful comments. The research presented in this paper was supported by the Office of Naval Research (ONR) and the NSF (Grant number: ECCS1462245, ECCS1533983, and ECCS1407712).

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

APPENDIX

THEORETICAL RESULTS

In order to prove Theorem 2, we use similar techniques to those used in the original GAN paper [11]. In what follows, we analyze the minimax game defined by:

$$\min_G \max_D \sum_{\mathbf{S} \in \{0,1\}^d} \mathbb{E}_{\mathbf{X} \sim \mathcal{P}_{\mathbf{X}}} [\mathbb{E}_{\tilde{\mathbf{X}} \sim \tilde{\mathcal{P}}_{\mathbf{X}}(\mathbf{X})} [\mathbf{S} \cdot \log(D((\mathbf{X}, \tilde{\mathbf{X}})_{\text{swap}(S)})) + (\mathbf{1} - \mathbf{S}) \cdot \log(1 - D((\mathbf{X}, \tilde{\mathbf{X}})_{\text{swap}(S)}))]]$$

(7)

where we have used the version of $\mathcal{L}_D$ given by equation (4) in the main manuscript (i.e. without hinting). The theoretical results that follow are proven only for this version of the loss, though we do believe that the theorem holds more generally for the hinting version of the loss - this is backed up by our empirical results demonstrating strict FDR control while using the hint mechanism. After proving that the optimal solution to this game does indeed provide us with valid knockoffs, we will then show that the additional term $\mathcal{L}_f$ does not change the optimal solution. Let $p$ be the density of $(\mathbf{X}, \tilde{\mathbf{X}})$.

We begin by stating a lemma, that follows from a similar proof to Proposition 1 in [11].

**Lemma 1.** *Let $(\mathbf{x}, \tilde{\mathbf{x}}) \in \mathcal{X} \times \mathcal{X}$. Then for a fixed generator, $G$, the $i^{th}$ component of the optimal discriminator, $D^*((\mathbf{x}, \tilde{\mathbf{x}}))$ is given by*

$$D^*((\mathbf{x}, \tilde{\mathbf{x}}))_i = \frac{p((\mathbf{x}, \tilde{\mathbf{x}})_{swap(\{i\})})}{p((\mathbf{x}, \tilde{\mathbf{x}})_{swap(\{i\})}) + p((\mathbf{x}, \tilde{\mathbf{x}}))}$$

(8)

*for each $i \in \{1, ..., d\}$.*

*Proof.* The proof of this involves some basic integral manipulation to get that the objective in (7) can be rewritten as

$$\sum_{i=1}^d \int_{\mathcal{X} \times \mathcal{X}} \log D((\mathbf{x}, \tilde{\mathbf{x}}))_i p((\mathbf{x}, \tilde{\mathbf{x}})_{\text{swap}(\{i\})}) + \log(1 - D((\mathbf{x}, \tilde{\mathbf{x}}))_i) p((\mathbf{x}, \tilde{\mathbf{x}})) \mathrm{d}\mathbf{x}.$$

We then observe that $y \mapsto a \log y + b \log(1 - y)$ achieves its maximum in $[0, 1]$ at $\frac{a}{a+b}$ and so the objective is maximized (with respect to $D$, for fixed $G$) when

$$D^*((\mathbf{x}, \tilde{\mathbf{x}}))_i = \frac{p((\mathbf{x}, \tilde{\mathbf{x}})_{\text{swap}(\{i\})})}{p((\mathbf{x}, \tilde{\mathbf{x}})_{\text{swap}(\{i\})}) + p((\mathbf{x}, \tilde{\mathbf{x}}))}$$

for each $i \in \{1, ..., d\}$. $\qquad\qquad\square$

With this lemma, we are now able to prove our key result.

**Theorem 3.** *Equation (7) is maximized (with respect to $G$) if and only if equation (1) (in the main paper) is satisfied by $G$.*

*Proof.* We begin by rewriting our loss, substituting in $D^*$, to give us the following loss for $G$:

$$\mathcal{L}_G = \sum_{S \subset \{1,...,d\}} \mathbb{E}_{\mathbf{X}, \tilde{\mathbf{X}}} \Big( \sum_{i \in S} \log \frac{p((\mathbf{x}, \tilde{\mathbf{x}})_{\text{swap}(S \setminus i)})}{p((\mathbf{x}, \tilde{\mathbf{x}})_{\text{swap}(S \setminus i)}) + p((\mathbf{x}, \tilde{\mathbf{x}})_{\text{swap}(S)})}$$

$$+ \sum_{i \notin S} \log \frac{p((\mathbf{x}, \tilde{\mathbf{x}})_{\text{swap}(S)})}{p((\mathbf{x}, \tilde{\mathbf{x}})_{\text{swap}(S \cup i)}) + p((\mathbf{x}, \tilde{\mathbf{x}})_{\text{swap}(S)})} \Big)$$

where we note that

$$((\mathbf{x}, \tilde{\mathbf{x}})_{\text{swap}(S)})_{\text{swap}(\{i\})} = \begin{cases} (\mathbf{x}, \tilde{\mathbf{x}})_{\text{swap}(S \setminus i)} & \text{if } i \in S \\ (\mathbf{x}, \tilde{\mathbf{x}})_{\text{swap}(S \cup i)} & \text{if } i \notin S. \end{cases}$$

Then by inspecting each term in the sum, we see that each term is a KL-divergence term that is minimized only when

$$p((\mathbf{x}, \tilde{\mathbf{x}})_{\text{swap}(\{S \setminus i\})}) = p((\mathbf{x}, \tilde{\mathbf{x}})_{\text{swap}(S)}) \qquad (9)$$

and

$$p((\mathbf{x}, \tilde{\mathbf{x}})_{\text{swap}(\{S \cup i\})}) = p((\mathbf{x}, \tilde{\mathbf{x}})_{\text{swap}(S)})$$

for every $i \in \{1, ..., d\}$, every $S \subset \{1, ..., d\}$ and each $(\mathbf{x}, \tilde{\mathbf{x}}) \in \mathcal{X} \times \mathcal{X}$.

By iteratively applying equation (9), we get that

$$p((\mathbf{x}, \tilde{\mathbf{x}})_{\text{swap}(S)}) = p((\mathbf{x}, \tilde{\mathbf{x}})_{\text{swap}(S \setminus 1)}) = ... = p((\mathbf{x}, \tilde{\mathbf{x}})_{\text{swap}(S \setminus \{1, ..., d-1\})})$$
$$= p((\mathbf{x}, \tilde{\mathbf{x}})_{\text{swap}(S \setminus \{1, ..., d\})}) = p((\mathbf{x}, \tilde{\mathbf{x}}))$$

proving the theorem. □

**Lemma 2.** *The addition of the term $\mathcal{L}_f$ to our loss, does not affect the optimal solution to it.*

*Proof.* By theorem 2, it suffices to show that any distribution satisfying equation (1) also minimizes $\max_f \mathcal{L}_f$. But we note that, as shown in [2], $\max_f \mathcal{L}_f$ (or rather $\sup_f \mathcal{L}_f$) is the Wasserstein distance between $\mathbf{X}$ and $\tilde{\mathbf{X}}$, which is 0 (and minimal) when $\mathbf{X} \overset{d.}{=} \tilde{\mathbf{X}}$. It therefore suffices to show that equation (1) implies $\mathbf{X} \overset{d.}{=} \tilde{\mathbf{X}}$.

Let $S = \{1, ..., d\}$. Then if $(\mathbf{X}, \tilde{\mathbf{X}})$ satisfy equation (1), we get that $(\mathbf{X}, \tilde{\mathbf{X}}) \overset{d.}{=} (\tilde{\mathbf{X}}, \mathbf{X})$. Since the joint distributions are equal, it follows that the marginal distributions are equal and so by projecting onto the first $d$ variables, we get that $\mathbf{X} \overset{d.}{=} \tilde{\mathbf{X}}$. □

## MINE

We state the key theory used by MINE to estimate the mutual information. For full details see the original paper, [4].

The mutual information is defined as

$$I(U; V) = \int_{\mathcal{U} \times \mathcal{V}} \log \frac{d\mathbb{P}_{UV}}{d\mathbb{P}_U \otimes d\mathbb{P}_V} d\mathbb{P}_{UV}$$

where $U$ and $V$ are random variables over some spaces $\mathcal{U}$ and $\mathcal{V}$, respectively with joint measure $\mathbb{P}_{UV}$ and marginal measures $\mathbb{P}_U = \int_{\mathcal{V}} d\mathbb{P}_{UV}$ and $\mathbb{P}_V = \int_{\mathcal{U}} d\mathbb{P}_{UV}$, respectively.

The mutual information can also be characterized by the Kullback-Leibler divergence, $D_{KL}$, as

$$I(U; V) = D_{KL}(\mathbb{P}_{UV} || \mathbb{P}_U \otimes \mathbb{P}_V)$$

The Donsker-Varadhan representation then gives us for any two probability measures $\mathbb{P}$ and $\mathbb{Q}$ over a probability space $\Omega$

$$D_{KL}(\mathbb{P} || \mathbb{Q}) = \sup_{T:\Omega \to \mathbb{R}} \mathbb{E}_{\mathbb{P}}[T] - \log(\mathbb{E}_{\mathbb{Q}}[e^T])$$

where the supremum is taken over all functions T such that the two expectations are finite.

A simple corollary of this is that fixing a class $\mathcal{F}$ of functions (such as a parametrized class $\{T_\theta : \theta \in \Theta\}$) over which the supremum is taken will provide us with a lower bound for the mutual information that approaches the true mutual information as the class becomes sufficiently rich. MINE [4] fix the class to be parametric in this way - given a fixed neural network architecture, they let $\mathcal{F} = \{T_\theta : \theta \in \Theta\}$ be the set of all functions parametrized by this network.

## IMPLEMENTATION OF KNOCKOFFGAN

In the experiments, the depth of the generator, discriminator and WGAN-GP networks is set to 4 and power network is set to 3. The number of hidden nodes in each layer is $d/4$, $d/16$ $d/4$ for the generator, discriminator, and WGAN-GP, respectively. For the power network, we use 2 diagonal matrices for each layer to make two hidden nodes for each feature separately. We use ReLu and tanh as the activation functions for each layer except for the output layer where we use a linear activation function for the generator, power network and WGAN-GP networks and sigmoid activation function for the discriminator network. The number of samples in each mini-batch is 128. KnockoffGAN is implemented in tensorflow.

## DETAILS OF BENCHMARKS

We use the following links for the implementations of 3 benchmarks.

- Knockoff: `http://web.stanford.edu/group/candes/knockoffs/software/knockoff/index.html`

- BHq Max: `http://web.stanford.edu/group/candes/knockoffs/software/knockoff/tutorial-4-r.html`

- BHq Marginal: Modifying the original code of BHq Max in `http://web.stanford.edu/group/candes/knockoffs/software/knockoff/tutorial-4-r.html`

Except for the knockoff generation step, KnockoffGAN follows the same procedures as the original knockoff framework described at `http://web.stanford.edu/group/candes/knockoffs/software/knockoff/tutorial-2-r.html` to select features.

ADDITIONAL EXPERIMENTS

INDEPENDENT GAUSSIANS

In the following experiment, features were taken to be i.i.d. Gaussian, with mean 0 and variance $\frac{1}{n}$, i.e. $\mathbf{X} \sim \mathcal{N}(0, \frac{1}{n}\mathbf{I}_n)$. The results are reported in Fig. 5.

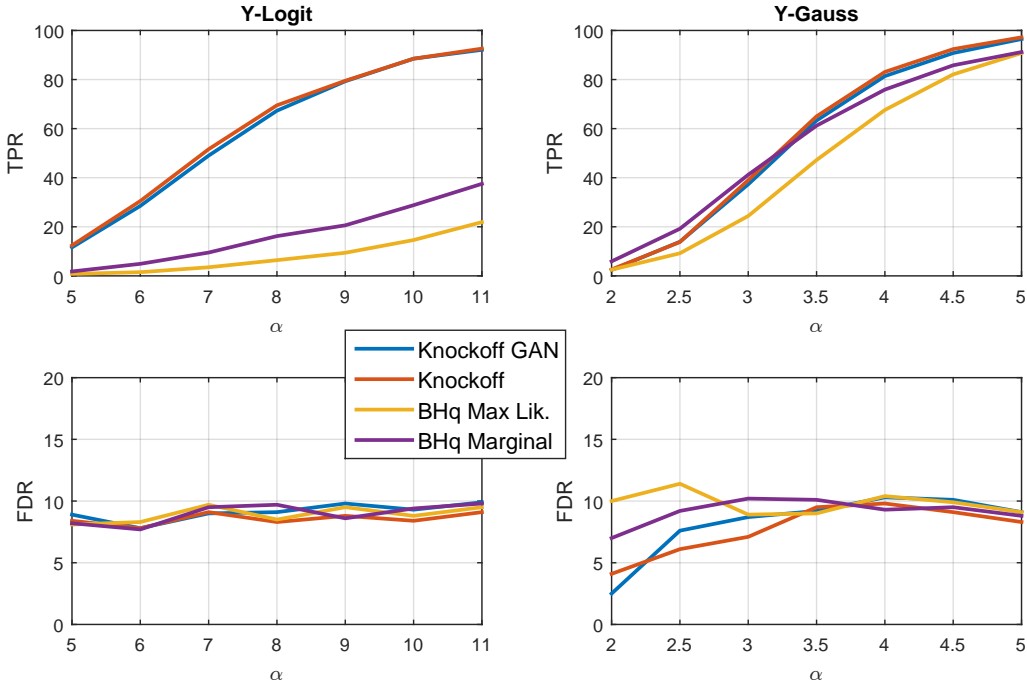

Figure 5: Comparison of KnockoffGAN with the benchmarks for $\mathbf{X} \sim \mathcal{N}(0, \frac{1}{n}\mathbf{I}_n)$. TPR is used to quantify performance and FDR is reported to verify that it is at the specified threshold (10%).

As we see in Fig. 5, all methods control the FDR at or very close to the specified 10% threshold. We also see that across the entire range of $\alpha$, KnockoffGAN achieves a very similar TPR to the original knockoff framework.

NON-GAUSSIAN SETTINGS

INDEPENDENT UNIFORM

In this experiment we set the feature distribution to be a Uniform distribution with mean $0$ and variance $\frac{1}{n}$ (to be consistent with the Gaussian experiments). Fig. 6 displays the results for each component of $\mathbf{X}$ being i.i.d. $\mathcal{U}(-\sqrt{3/n}, \sqrt{3/n})$. Once again we see that KnockoffGAN consistently outperforms the original knockoff framework, achieving a higher TPR across the entire range of $\alpha$ in both settings.

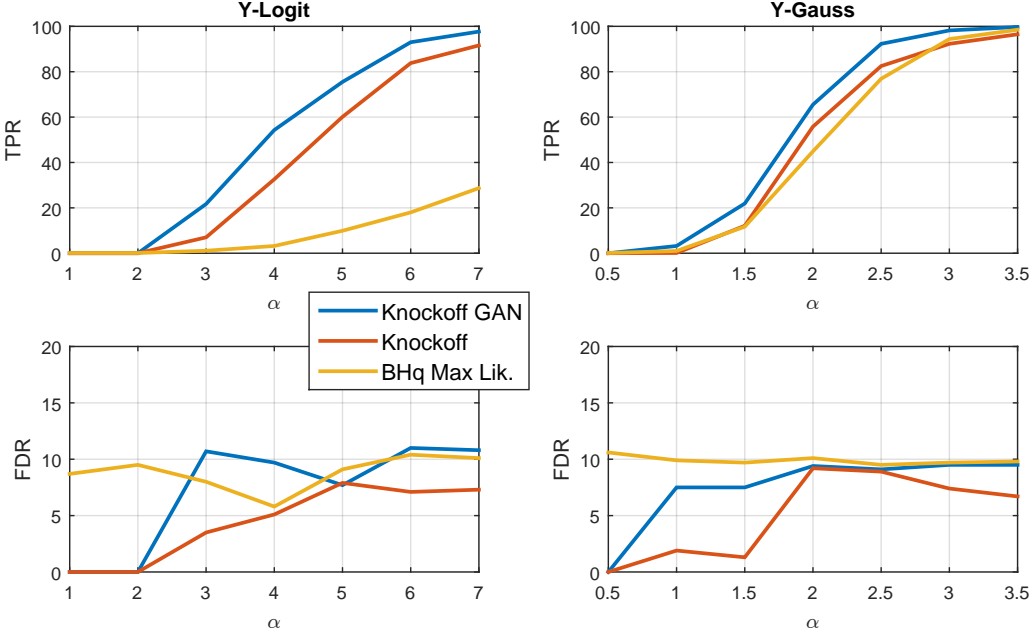

Figure 6: Comparison of KnockoffGAN with the benchmarks for $\mathbf{X} \sim \mathcal{U}(-\sqrt{3/n}, \sqrt{3/n})$. TPR is used to quantify performance and FDR is reported to verify that it is at the specified threshold (10%).

DIRICHLET

In this experiment we set the feature distribution to be a Dirichlet$(1, ..., 1)$ distribution - i.e. the uniform distribution over the $(d-1)$-simplex. Correlation here exists through the requirement that $\sum_{i=1}^{d} X_i = 1$.

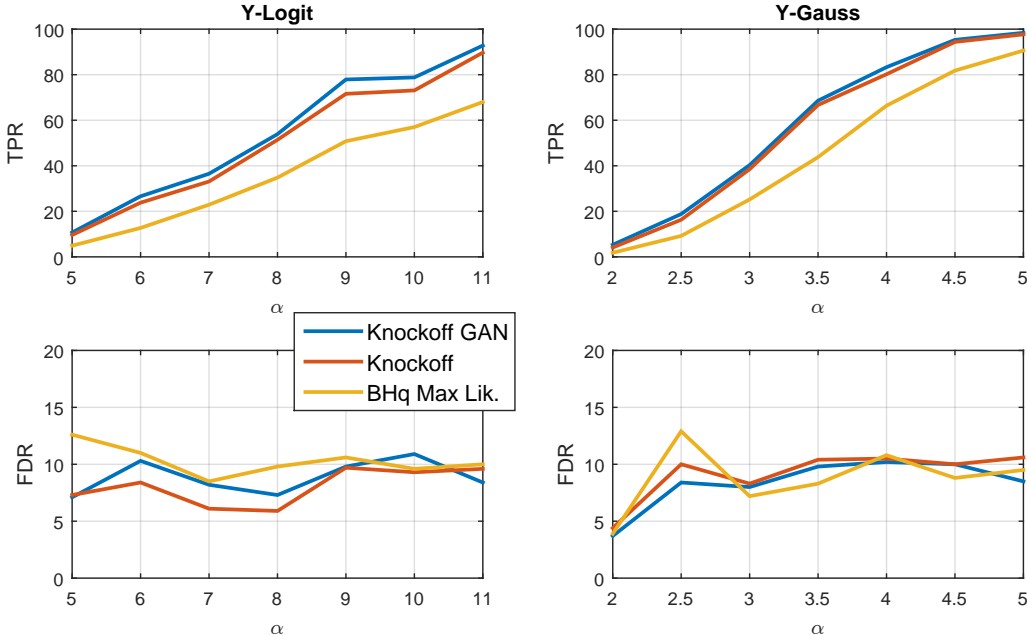

Figure 7: Comparison of KnockoffGAN with the benchmarks for $\mathbf{X} \sim$ Dirichlet$(1, ..., 1)$. TPR is used to quantify performance and FDR is reported to verify that it is at the specified threshold (10%).

GAUSSIAN MIXTURE MODELS

For the GMM2 model we set the means $(\mathbf{m}^1, \mathbf{m}^2)$ of the 2 Gaussians to be:

- $m_i^1 = 1$ for $i = 1$ to 100 and 0 for $i = 101$ to 1000,
- $m_i^2 = -1$ for $i = 1$ to 100 and 0 for $i = 101$ to 1000.

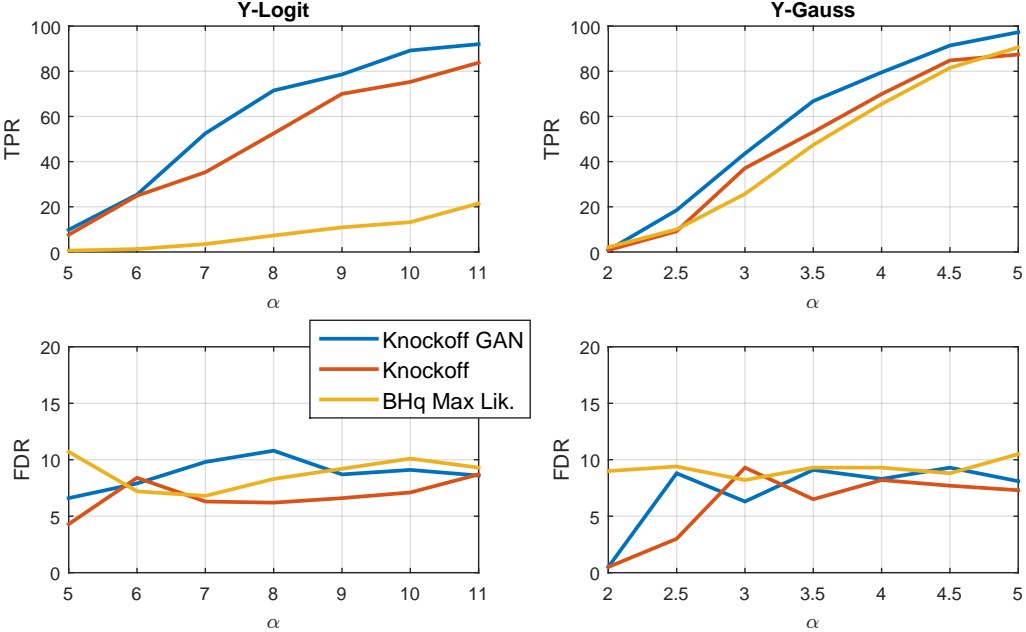

Figure 8: Comparison of KnockoffGAN with the benchmarks for $\mathbf{X} \sim$ GMM2. TPR is used to quantify performance and FDR is reported to verify that it is at the specified threshold (10%).

For the GMM3 model we set the means $(\mathbf{m}^1, \mathbf{m}^2, \mathbf{m}^3)$ of the 3 Gaussians to be:

- $m_i^1 = 1$ for $i = 1$ to 100 and 0 for $i = 101$ to 1000,
- $m_i^2 = 1$ for $i = 1$ to 50 and -1 for $i = 51$ to 100 and 0 for $i = 101$ to 1000,
- $m_i^3 = -1$ for $i = 1$ to 100 and 0 for $i = 101$ to 1000.

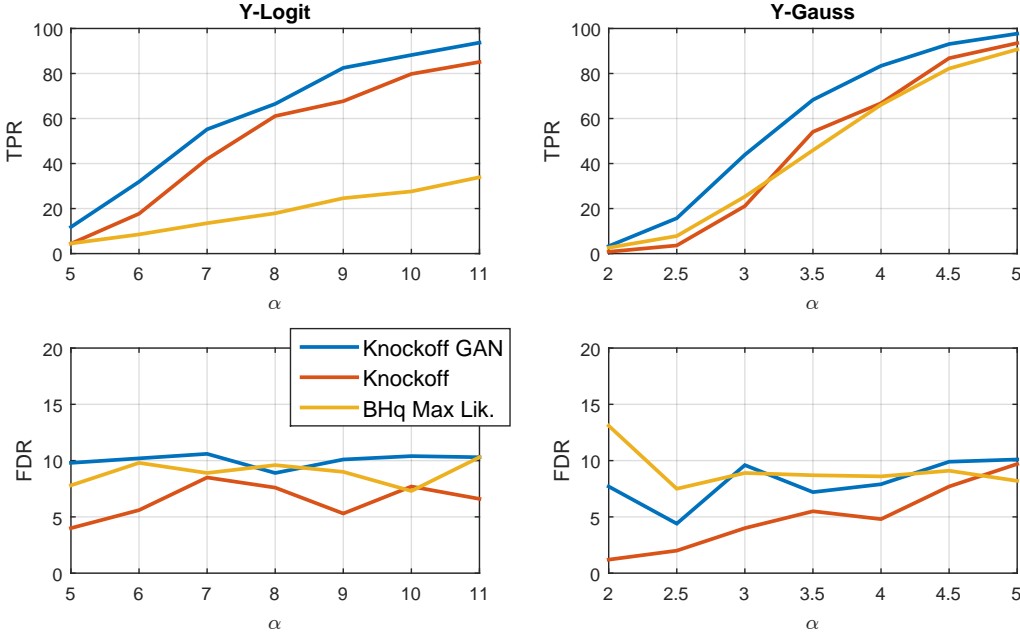

Figure 9: Comparison of KnockoffGAN with the benchmarks for $\mathbf{X} \sim$ GMM3. TPR is used to quantify performance and FDR is reported to verify that it is at the specified threshold (10%).

## HYPER-PARAMETER ANALYSIS

Hyper-parameter selection in the feature selection problem is difficult; hyper-parameter selection cannot be performed using cross-validation as we do not have access to ground truth. The hyper-parameters must therefore be fixed a priori. For this we believe that $\lambda = 1$ and $\mu = 1$ are perhaps the most canonical choice we could make and thus these were used in the main manuscript. In the following experiments, we investigate the sensitivity of the results to various settings of $\lambda$ and $\mu$. The results below are in the auto-regressive Uniform setting with Y-Logit and FDR set to 10%.

| TPR | | $\lambda$ | | | | | |
|---|---|---|---|---|---|---|---|
| | | 0 | 0.1 | 0.5 | 1 | 5 | 10 |
| | 0.1 | 70.3 | 76.2 | 75.7 | 77.0 | 75.1 | 76.0 |
| $\phi$ | 0.2 | 68.7 | 72.6 | 71.9 | 72.7 | 72.9 | 73.7 |
| | 0.4 | 42.4 | 55.1 | 51.1 | 53.1 | 53.8 | 53.2 |
| | 0.8 | 22.4 | 27.4 | 25.3 | 27.7 | 27.5 | 27.6 |

Table 2: Evaluation of the hyper-parameter $\lambda$ in the auto-regressive uniform setting with Y-Logit and FDR threshold 10%

| TPR | | $\mu$ | | | | | |
|---|---|---|---|---|---|---|---|
| | | 0 | 0.1 | 0.5 | 1 | 5 | 10 |
| | 0.1 | 70.3 | 76.2 | 75.7 | 77.0 | 75.1 | 76.0 |
| $\phi$ | 0.2 | 68.7 | 72.6 | 71.9 | 72.7 | 72.9 | 73.7 |
| | 0.4 | 42.4 | 55.1 | 51.1 | 53.1 | 53.8 | 53.2 |
| | 0.8 | 22.4 | 27.4 | 25.3 | 27.7 | 27.5 | 27.6 |

Table 3: Evaluation of the hyper-parameter $\mu$ in the auto-regressive uniform setting with Y-Logit and FDR threshold 10%

As can be seen in Table 2 and 3, the performance of KnockoffGAN is not sensitive to the value of $\lambda$ and $\mu$. The only significant difference can be seen when either $\lambda$ or $\mu$ is set to 0 which represents the exclusion of either the power network or WGAN-discriminator network, respectively, from the model. In particular, the lack of sensitivity to $\mu$ aligns with Lemma 2 in which we see that there is not a trade-off between $\mathcal{L}_f$ and $\mathcal{L}_D$ but rather the two can be simultaneously minimised.

To further understand the effects of the WGAN network, we report the final values of the other losses ($\mathcal{L}_D$ and $\mathcal{L}_P$) when $\mu = 1$ and $\mu = 0$ (i.e. with and without the WGAN regularisation). The results are given below in the auto-regressive Uniform setting with Y-Logit and FDR set to 10%.

| Loss | | $\mathcal{L}_D$ | | $\mathcal{L}_P$ | |
|---|---|---|---|---|---|
| | | $\mu = 0$ | $\mu = 1$ | $\mu = 0$ | $\mu = 1$ |
| | 0.1 | 0.6894 | 0.6962 | 0.0014 | 0.0203 |
| $\phi$ | 0.2 | 0.7005 | 0.6964 | 0.0018 | 0.0115 |
| | 0.4 | 0.6919 | 0.6955 | 0.0046 | 0.0180 |
| | 0.8 | 0.7013 | 0.6960 | 0.0068 | 0.0311 |

Table 4: Final values of the other training losses when $\mu = 0$ or $\mu = 1$ in the auto-regressive uniform setting with Y-Logit and FDR threshold 10%

As can be seen in Table 4, the inclusion of the WGAN network improves the final value of $\mathcal{L}_D$, bringing it significantly closer to its optimal value of $\log(2) \approx 0.693$. On the other hand, the power network loss is increased, however, this trade-off is expected and acceptable - the FDR guarantees rely on the discriminator loss ($\mathcal{L}_D$) and not the power network. As we see in Table 3, the TPR does not suffer from this increased loss for the power network.

For our final hyper-parameter evaluation, we investigate the effect of varying the hinting probability, $p$, which determines the probability with which $B_i = 1$. We note that this hyper-parameter is used to trade-off between speed of learning and optimality of the learned solution. A low probability makes for fast convergence, but suboptimal convergence whereas a high probability makes for slow convergence (but a more optimal solution). We chose $0.9$ to balance this, following the implementation of [38]. To demonstrate this trade-off we vary this hyper-parameter (from 0 to 0.9). The results below are for the auto-regressive Uniform distribution with Y-Logit and FDR set to 10%.

| TPR | | $p$ | | | | | |
|-----|-----|------|------|------|------|------|------|
| | | 0 | 0.2 | 0.4 | 0.6 | 0.8 | 0.9 |
| | 0.1 | 73.3 | 74.7 | 75.0 | 75.3 | 75.7 | 77.0 |
| | 0.2 | 67.7 | 68.4 | 69.1 | 69.7 | 72.4 | 72.7 |
| $\phi$ | 0.4 | 51.2 | 51.6 | 51.7 | 51.9 | 52.9 | 53.1 |
| | 0.8 | 18.0 | 20.4 | 21.4 | 22.4 | 25.4 | 27.7 |

Table 5: Evaluation of the hinting probability, $p$, in the auto-regressive uniform setting with Y-Logit and FDR threshold 10%

## RESULTS WITH FDR THRESHOLD 5%

In all of the synthetic experiments above, the FDR threshold is set to 10% (which is the level most thoroughly investigated by [3]). In this final experiment, we set the FDR threshold to 5% (which aligns with our real-world experiment) and verify that KnockoffGAN is capable of controlling the FDR at this level and still outperforms the original knockoff framework. We conduct the experiments on (1) an auto-regressive distribution with Gaussian marginal distributions, (2) an auto-regressive distribution with Uniform marginal distributions, and (3) a 4-mixture Gaussian mixture model.

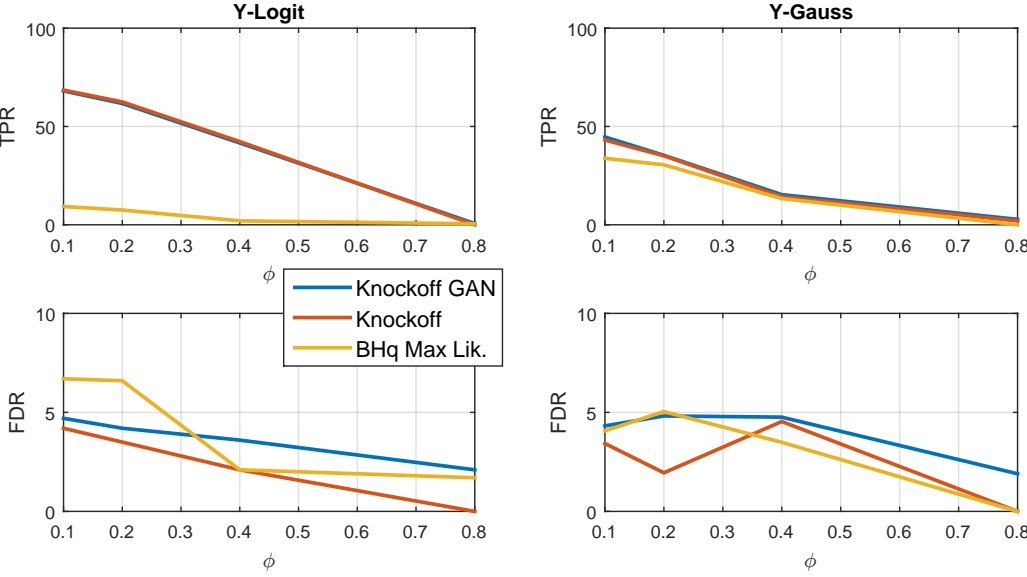

Figure 10: A comparison of the performance of KnockoffGAN for $\mathbf{X}$ distributed as an auto-regressive distribution with Gaussian marginal distributions. TPR is used to quantify performance and FDR is reported to verify that it is at the specified threshold (5%).

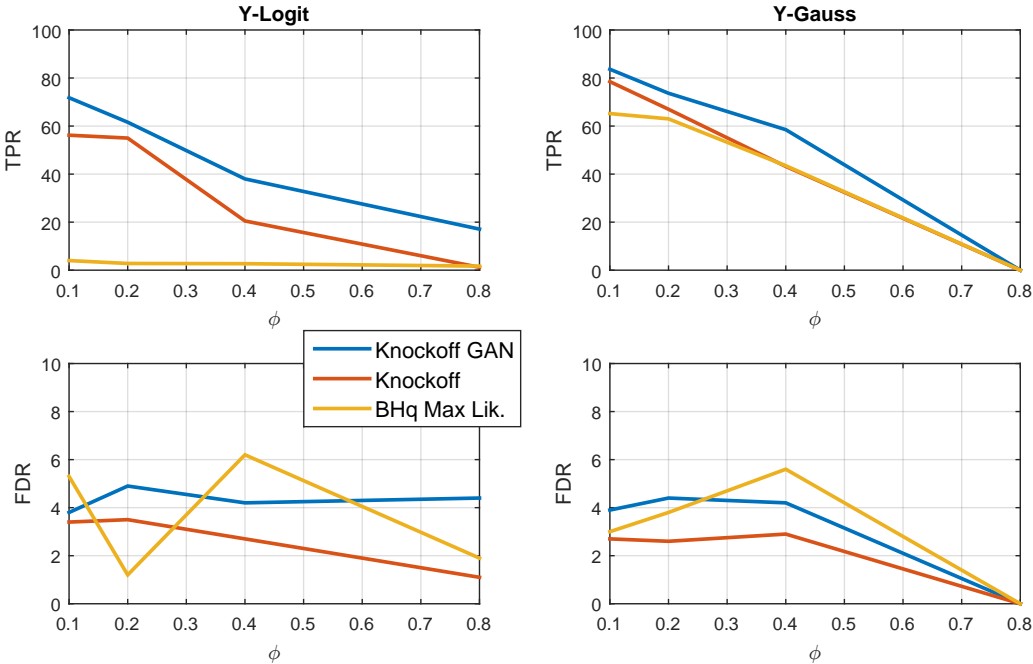

Figure 11: A comparison of the performance of KnockoffGAN for **X** distributed as an auto-regressive distribution with Uniform marginal distributions. TPR is used to quantify performance and FDR is reported to verify that it is at the specified threshold (5%).

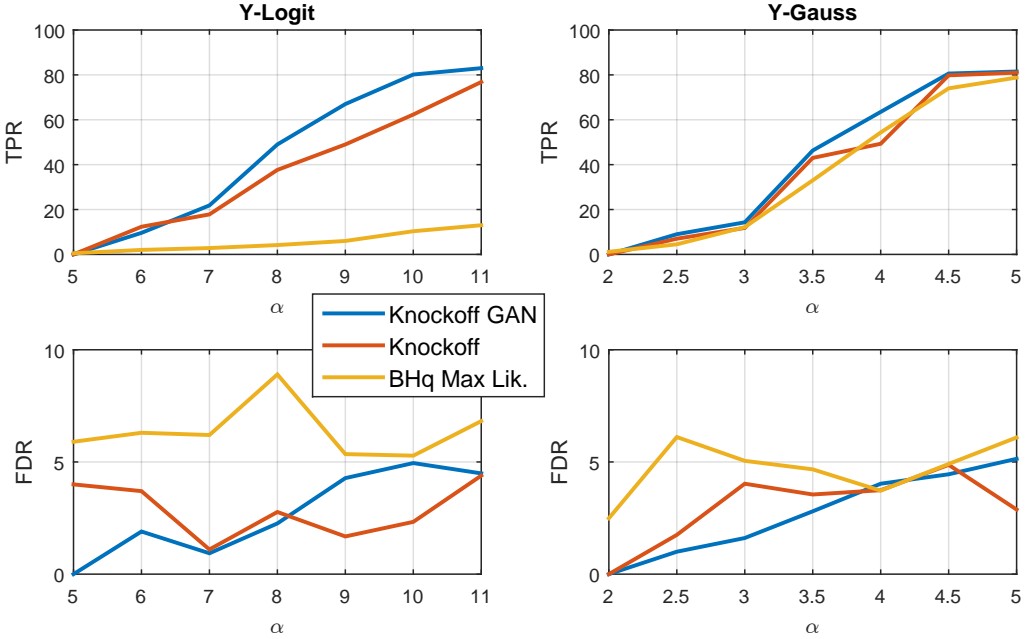

Figure 12: A comparison of the performance of KnockoffGAN for **X** distributed as 4-mixture Gaussian mixture model. TPR is used to quantify performance and FDR is reported to verify that it is at the specified threshold (5%).

## STATISTICS OF REAL-WORLD BIOBANK DATASET

To preserve anonymity of the authors, the full details of this dataset will be given upon acceptance of the paper. In Table 1, we provide the basic statistics of the real-world biobank dataset.

| Staticstics | Values |
|---|---|
| No of patients | **86082** |
| No of features | **387** |
| Pearson correlation across the features | 25%: **0.0042**, 50%: **0.0120** ,75%: **0.0332** |
| Pearson correlation with the outcome | **CVD:** 25%: **0.0033**, 50%: **0.0091** ,75%: **0.0199** 
 **Diabetes:** 25%: **0.0068**, 50%: **0.0179** ,75%: **0.0439** |
| Label distribution | **CVD:** 1252 patients (1.5%) 
 **Diabetes:** 3932 patients (4.6%) |

Table 6: Basic statistics of real-world biobank dataset (% means percentile in Pearson correlation rows)

