# OpenReview forum: "KnockoffGAN: Generating Knockoffs for Feature Selection using Generative Adversarial Networks"
_ICLR.cc/2019/Conference_

### Official Review · AnonReviewer2 · 2018-10-30

**Rating:** 7
**Confidence:** 4

**Review:**

This paper introduces a novel feature selection method by utilizing GAN to learn the distributions. The novelty of this paper is to incorporate two recent works, i.e. knockoff for feature selection and W-GAN for generative models. Compared to the latest knockoff work which requires a known multivariate Gaussian distribution for the feature distribution, the proposed work is able to generate knockoffs for any distribution and without any prior knowledge of it.

Pros: This paper is very well written. I enjoyed reading this paper. It is novel and addresses an important problem. The numerical study clearly shows the advantage of the proposed work.

Cons:

Q1: In the discriminator, instead of training with respect to the full loss, the authors consider to mask some information by using a multivariate Bernoulli random variable $B$ with success probability 0.9. Then the discriminator needs to predict only when $B_i = 0$. Can the authors provide some justification of such choice of the parameters? This choice is a little bit mysterious to me.

Q2: How sensitive are the hyper-parameters $\eta$ (set to 10 in the experiments), $\lambda$, and $\mu$ (set to 1 in the experiments)?

Q3: In the real data example, the feature selection performance is less justified as there is no truth. One suggestion is to evaluate the prediction errors using the selected features and compare with the benchmarks.

---

> ### Author Response · Authors · 2018-11-14
> **RE: review**
>
> Thank you for your insightful comments.
>
> A1: We would first like to note that selecting hyper-parameters in this setting is not possible through conventional means, and they typically need to be selected in advance as we do not have access to any ground truth that allows us to perform cross-validation for hyper-parameter optimization.
> As for this hyper-parameter (B_i) in particular, we note that this hyper-parameter is used to trade-off between speed of learning and optimality of the learned solution. A low probability makes for fast convergence, but suboptimal convergence whereas a high probability makes for slow convergence (but a “more” optimal solution). We chose 0.9 to balance this, following the implementation of [36]. To demonstrate this trade-off we will include results for our method in which we vary this hyper-parameter (from 0 to 0.9) in the revised manuscript. The results below are for the auto-regressive Uniform distribution with Y-Logit, FDR set to 10% for various values of the success probability of B_i:
> ---------------------------------------------------------------------------------------------
> Phi / B_i  |      0      |     0.2     |     0.4     |     0.6     |     0.8     |     0.9     |
> ---------------------------------------------------------------------------------------------
>      0.1      |    73.3   |    74.7    |    75.0    |    75.3    |   75.7     |   77.0     |
>      0.2      |    67.7   |    68.4    |    69.1    |    69.7    |   72.4     |   72.7     |
>      0.4      |    51.2   |    51.6    |    51.7    |    51.9    |   52.9     |   53.1     |
>      0.8      |    18.0   |    51.7    |    21.4    |    22.4    |   25.4     |   27.7     |
> ---------------------------------------------------------------------------------------------
>
> A2: As noted above it is not possible to tune hyperparameters in this setting as we do not have access to any ground truth; however, we will include results for various hyper-parameter settings (various lambda and mu) to evaluate their sensitivity.
> Setting eta = 10 is standard practice in the WGAN literature; thus, we fixed it. Finally, we would like to reassure you that we did not cherry pick these hyperparameters (lambda and mu) - they were set to 1 which is perhaps the most “canonical” choice we could have made.
> In the below, we give the results for varying lambda in the auto-regressive Uniform setting with Y-Logit and FDR set to 10%:
> ---------------------------------------------------------------------------------------------------
> Phi / Lambda |      0      |     0.1     |     0.5     |       1       |      5       |     10      |
> ---------------------------------------------------------------------------------------------------
>               0.1      |    70.3   |    76.2    |    75.7    |    77.0    |   75.1     |   76.0     |
>               0.2      |    68.7   |    72.6    |    71.9    |    72.7    |   72.9     |   73.7     |
>               0.4      |    42.4   |    55.1    |    51.1    |    53.1    |   53.8     |   53.2     |
>               0.8      |    22.4   |    27.4    |    25.3    |    27.7    |   27.5     |   27.6     |
> ---------------------------------------------------------------------------------------------------
> Phi / mu          |      0      |     0.1     |     0.5     |       1       |      5       |     10      |
> ---------------------------------------------------------------------------------------------------
>               0.1      |    70.0   |    79.3    |    79.0    |    77.0    |   77.3     |   79.0     |
>               0.2      |    63.4   |    74.4    |    73.4    |    72.7    |   74.0     |   73.7     |
>               0.4      |    42.1   |    49.8    |    50.1    |    53.1    |   52.8     |   50.8     |
>               0.8      |    22.0   |    27.7    |    29.7    |    27.7    |   29.4     |   23.4     |
> ---------------------------------------------------------------------------------------------------
>
> A3: We agree that the real data does not provide strong evidence for the efficacy of our method, and would point to the synthetic data as the main source of evidence. However, we performed the real data experiment as a qualitative experiment, hoping to demonstrate that at the very least the method is capable of discovering known relevant features (according to PubMed).
> While we could use predictive power of the selected features as a metric, we do not believe this would be a meaningful metric here. The focus of this method (and Knockoffs in general) is on discovery of relevant variables and not on selecting variables for prediction. The predictive power of a set of features, while (most likely) correlated with TPR, will not necessarily increase monotonically with TPR.

---

### Official Review · AnonReviewer3 · 2018-10-31
**Compelling approach to an important problem in FDR-controlled feature selection, with only minor weaknesses.**

**Rating:** 10
**Confidence:** 4

**Review:**

This manuscript describes an extension of the knockoff framework, which is designed to carry out feature selection while controlling the FDR among selected features, to settings in which the generative distribution of the features is not Gaussian. Specifically, the authors employ a GAN (with several modifications and additions) in which the generator produces knockoffs and the discriminator attempts to identify which features have been swapped between the original and the knockoff.

The method works as follows: (1) A conditional generator takes random noise and the real features as input, and outputs knockoff features. (2) A modified discriminator is used in such a way that the generator learns to generate knockoffs satisfying the necessary swap condition, so as to control the FDR of the knockoff procedure. (3) A power network uses Mutual Information Neural Estimation (MINE) to estimate the mutual information between each feature and its knockoff counterpart, so as to maximize the power of the knockoff procedure.

Results are provided on synthetic and real data. As for the synthetic data, when the underlying feature distribution is Gaussian, the proposed method, KnockoffGAN, performs almost as well as the original knockoff and outperforms the BHq method; when the underlying feature distribution is non-Gaussian, KnockoffGAN dominates both the original knockoff and BHq methods. As for the real data, the authors claim to identify nine relevant features for cardiovascular disease and eight relevant features for diabetes, whereas the original knockoff procedure identifies zero features from the same data.

General comments:

This is an extremely impressive piece of work. The manuscript itself is a pleasure to read, and the results clearly demonstrate that the proposed KnockoffGAN both controls FDR and achieves power comparable to the original knockoff procedure in the Gaussian setting and much better than the original knockoff when the underlying distribution is not Gaussian.

Strengths:

The combination of GANs and knockoff filter is a very promising and intriguing idea.

The use of the modified discriminator to ensure that the generated knockoffs satisfy the necessary swap condition is novel and intuitively sound.

The use of MINE to maximize power by maximizing the mutual information between each feature and its knockoff counterpart is also interesting.

The paper is well written, reads smoothly and the ideas are well exposed.

The illustrative figure is straightforward.

Weaknesses:

Intuitively, the modified discriminator and the power network should conflict with each other. I expect it was tricky to achieve a good tradeoff between two, but the authors failed to elaborate on these details.

The authors do not provide the design details of the neural networks. How dependent on the specific parametrization of the network architecture is the performance? How does the training order of four networks matter to the performance?

The manuscript should cite [[ Jaime Roquero Gimenez, Amirata Ghorbani, and James Zou. "Knockoffs   for the mass: new feature importance statistics with false discovery   guarantees." arXiv:1807.06214, 2018. ]] which proposes a way to generate knockoffs for a Gaussian mixture model, and this method should be included in the relevant supplementary figure.

In Section 5.1.4, I would like to know, for a fixed data set, how the regularization affects the final values of the other loss terms.

The analysis of real data in Section 5.2 is unsatisfying in several respects.

First, there is an unfortunate oversight in Table 1: the text refers to three features that are "trivial," but only one of these is marked with an asterisk.  This leaves open the question of whether there are other trivial features beyond the three mentioned in the text. In addition, it is not clear exactly what it means for a feature to be "trivial" in this context.

This point gets to a deeper problem with the evaluation, which is that we are told, with no evidence, that these features are supported by literature in PubMed.  I would like to see two things here.  First, it seems obvious to me that if you are going to say that there is support in PubMed, you are obliged to actually report the citations that supposedly give this support. This could be done in the appendix. Equally importantly, there is a potential here for ascertainment bias which should be combatted in some fashion.  Presumably, some human expert had to do the PubMed searches to make this assessment.  I would like to know how "permissive" this assessor is.  To assess this, one could give the assessor a collection of terms, some of which were selected by KnockoffGAN and some at random, and then report the results. Obviously, some features that are significant may not be in the list of selected terms (because KnockoffGAN does not achieve 100% power) and so may appear as false negatives. But without some assessment like this, I have trouble believing this assessment.

A related point is that it seems quite unfortunate that the authors chose a data set that cannot be described at all due to the anonymity constraint.  At the very least, it seems that we should be told the dimensionality of the data set. The Knockoff literature contains real data sets that could have been used here.

Minor comments:

On the first page, the sentence beginning "On the other hand," should clarify that this is only in expectation.

p. 3: Missing right paren after [7].

p. 5: Write out “Gaussian process.”

p. 5: "as little" -> "as little as possible"

p. 6: "to show that in" -> "to show, in"

In Figures 2-5, add a horizontal line at 10% FDR for reference.

p. 10: "features ones" -> "features"

Note to program committee:

I did not review the technical details of the proof in the appendix.

---

> ### Author Response · Authors · 2018-11-14
> **RE2: Compelling approach to an important problem in FDR-controlled feature selection, with only minor weaknesses.**
>
>
> A5: This was indeed an oversight and we will correct the text. We will change “trivial” to “well-known” and hopefully that will make clearer our point. The asterisk will be removed as we do not feel it helps provide any clarity.
>
> A6: The citations found in Table 1 are in fact citations to relevant PubMed literature who describe associations related to those found by our analysis. We will make this clearer by adding two columns to the table, each one containing the citations for the corresponding features.
> We certainly concede that this type of evaluation is qualitative at best, but is the best we can do on real data given that we don’t have access to the ground truth. We would point to the synthetic data experiments as the main source of evidence for the efficacy of our method.
>
> A7: For reference, the dataset we used is 387-dimensional in the real-world experiments. Of course, upon an acceptance decision we will be able to disclose more complete details of the dataset which will be included in the paper.
>
> A8: We will correct these, thank you.

---

> ### Author Response · Authors · 2018-11-14
> **RE1: Compelling approach to an important problem in FDR-controlled feature selection, with only minor weaknesses.**
>
> Thank you for your insightful comments.
>
> A1: Thank you for your insightful comments. Actually, the power network and modified discriminator do not directly conflict - the modified discriminator requires that a feature and its knockoff have the same conditional distribution given the other features, but can be independent otherwise. This “room for independence” is what the power network will be minimizing. Of course, this is not the same as being able to be fully independent and so there is some trade-off, but we found that in practice setting the trade-off parameter (lambda) to 1 worked well.
> It should be noted that hyper-parameter selection cannot be performed using cross-validation as we do not have access to ground truth and so the hyperparameters must be fixed a priori. For this we believe that 1 is a natural choice and as noted performed well in practice. Below are the results of varying lambda in the Y-Logit setting with features distributed as auto-regressive uniform.
> ---------------------------------------------------------------------------------------------------
> Phi / Lambda |      0      |     0.1     |     0.5     |       1       |      5       |     10      |
> ---------------------------------------------------------------------------------------------------
>               0.1      |    70.3   |    76.2    |    75.7    |    77.0    |   75.1     |   76.0     |
>               0.2      |    68.7   |    72.6    |    71.9    |    72.7    |   72.9     |   73.7     |
>               0.4      |    42.4   |    55.1    |    51.1    |    53.1    |   53.8     |   53.2     |
>               0.8      |    22.4   |    27.4    |    25.3    |    27.7    |   27.5     |   27.6     |
> ---------------------------------------------------------------------------------------------------
>
> A2: Specific design details can be found in the Appendix under the “Implementation of KnockoffGAN” section. The training of the power, discriminator and WGAN discriminator is in fact independent (i.e. the updates of each of these do not depend on the weights of the other networks), and as such these 3 can be trained in any order (or even in parallel). Following standard practices in the WGAN literature we trained both discriminators and the power network for 5 iterations per generator iteration.
>
> A3: We were not aware of this paper, thank you. While this method does propose a more general method for generating knockoffs, it still has the same limitations of the original knockoff framework in that it is required to know that the distribution is from the class of distributions discussed in the paper (which includes Gaussian Mixture Model (GMM)). We would also note that, although we do perform experiments in a GMM setting, this was not a cherry-picked distribution, and the main goal here was to demonstrate results on non-Gaussian distributions for comparison with the original knockoff framework. This, we believe, demonstrates the more general result that when the distributions are mis-specified, the knockoff method struggles to give good performance, and this would apply to the above-mentioned paper in non-GMM settings as well. We will revise the manuscript to include a citation of this paper in the related works section and a brief discussion.
>
> A4: We will include the results of the regularization affects (mu = 0 and mu = 1) in terms of the final values of the other loss terms (L_D and L_P) in the revised supplementary materials. For easy reference, the results are given below for the auto-regressive uniform case with Y-Logit:
> --------------------------------------------------------------------------------------------
>                  |                    L_D                       |                       L_P                    |
>     Phi       |    Mu = 0       |     Mu = 1       |    Mu = 0       |     Mu = 1       |
> --------------------------------------------------------------------------------------------
>     0.1       |       0.6894     |     0.6962        |    0.0014       |     0.0203       |
>     0.2       |       0.7005     |     0.6964        |    0.0018       |     0.0115       |
>     0.1       |       0.6919     |     0.6955        |    0.0046       |     0.0180       |
>     0.1       |       0.7013     |     0.6960        |    0.0068       |     0.0311       |
> --------------------------------------------------------------------------------------------

---

### Official Review · AnonReviewer1 · 2018-11-12
**Good paper with limitations in empirical evaluation**

**Rating:** 6
**Confidence:** 4

**Review:**

The paper presents a deep-learning-based version of the knockoff method by Candes et al. for FDR control in feature selection problems to avoid assumptions posed on the distribution of features by the original method. In a supervised feature selection setting, the goal of the knockoff framework is to select a set of input features that are statistically associated to an output variable Y, while controling the FDR. The basic idea behind knockoff is to generate artificial input feature vectors, (i.e. knockoffs) that are independent of Y, when conditioned on the real feature vector X, but after swapping arbitrary elements with X, are distributed as X. Sets of associated features and FDR estimates are obtained by contrasting suited feature selction criteria that measure associations of knockoffs and real features with the target Y.
Lasso coefficients and random forests are used in the paper.
The main contribution of the current paper is the use of a GAN to generate knockoffs, and, according to the paper in particular, the use of a discriminator that tries to identify the positions of knockoff features that have been swapped into real feature vectors X to control equlaity in distribution between knockoffs and feature vectors. Additionally, a Wasserstein discriminator and a MINE loss are used to control the knockoff distribution. Otherwise, the paper follows the standard knockoff procedure.

The approach is evaluated in simulations, varying two degrees of freedom: i) Gaussian distribute features vs. features that follow mixtures of Gaussians. ii) Gaussian and logit distributions of Y conditioned on linear functions of a subset of X features.
Using a Lasso-based feature selection criterion, the GAN knockoff method achieves the highest TP rates among a number of methods that empirically are shown to roughly control a target FDR of 10%. However, the figures are too small to judge FDR control more fine-graoned than 10% +-5%. Here, I would have wished i) a higher resolution to demonstrate FDR control, as well as an evaluation of different FDR cutoffs, especially including smaller cutoffs.

Additionally, an appliction to real data is performed. However, this evaluation is not very informative for several reasons. i) The dataset is not specified, making the experiment intransparent and non-reproducible. ii) A different feature selection criterion based on random forests is used, compared to the Lasso-based criterion in the synthetic experiments. iii) A different FDR cutoff of 5% has been used compared to the simulations. It is not clear, if the method shows FDR control in synthetic settings at 5%. For these reasons, the real-world experiment is hardly comparable to the synthetic settings.


The paper is relatively well-written and clear. Discussion of related work is appropriate.

In sum, the paper has some limitations in the empirical evaluation, but nonetheless the use of a GAN promises significant gains in statistical power.

---

> ### Author Response · Authors · 2018-11-14
> **RE: Good paper with limitations in empirical evaluation**
>
> Thank you for the insightful comments.
>
> A1: We would first like to note that our synthetic experiments consist of Gaussian distributed features vs non-Gaussian distributed features (which include uniform, auto-regressive uniform and Dirichlet as well as mixtures of Gaussian). To make the FDR control clearer we will switch the FDR graphs from 0 to 100% scale to 0 to 20% scale in the revised manuscript. Third, we will add additional experiments with the FDR threshold set to 5% to align with the real world experiments (we note that FDR was set to 5% in the real data experiment so that the list of discovered true positives was manageable for cross-reference with PubMed). Below we give the results for autoregressive Uniform distribution in the Y-Logit setting at 5% FDR, further results will be added to the revised manuscript.
>
> -----------------------------------------------------------------------------------
> Phi/Methds (TPR) | KnockoffGAN | Knockoff | BHq Max Lik |
> -----------------------------------------------------------------------------------
>            0.1                 |         0.718       |     0.562    |     0.040          |
>            0.2                 |         0.616       |     0.550    |     0.028          |
>            0.4                 |         0.380       |     0.205    |     0.027          |
>            0.8                 |         0.171       |     0.012    |     0.017          |
> ------------------------------------------------------------------------------------
> Phi/Methds (FDR) | KnockoffGAN | Knockoff | BHq Max Lik |
> -----------------------------------------------------------------------------------
>            0.1                 |         0.038       |     0.034    |     0.053          |
>            0.2                 |         0.049       |     0.035    |     0.012          |
>            0.4                 |         0.042       |     0.027    |     0.062          |
>            0.8                 |         0.044       |     0.011    |     0.019          |
> ------------------------------------------------------------------------------------
>
> A2: i) The dataset details are not given at this time as we wish to remain anonymous for the review process. Upon an acceptance decision, we will include all details of the dataset in the paper.
> (ii) Our key contribution in this work is in having provided a new method for generating the knockoffs, and both the Lasso-based and Random Forest-based statistics we used are from existing knockoff works. In the synthetic data settings, the relationships are known to be linear in both the Y-Logit and Y-Gaussian settings, while in the real data setting the relationships are unknown. For this reason it makes sense to use the Lasso-based statistic in the synthetic data and the non-parametric RF-based estimator in real data.
> (iii) The results above demonstrate FDR control at 5% and further results will be added to the main manuscript (on more distributions than the AR-uniform given above).

---

### Public Comment · (anonymous) · 2019-09-25
**Knockoffs**

The discovery of relevant features is really important for performing a particular task.
https://spanishdictionary.cc/

---

### Meta-Review · Area_Chair1 · 2018-12-13

**Confidence:** 4
**Recommendation:** Accept (Oral)

**Metareview:**

The paper presents a novel strategy for statistically motivated feature selection i.e. aimed at controlling the false discovery rate. This is achieved by extending knockoffs to complex predictive models and complex distributions via (multiple) generative adversarial networks.

The reviewers and ACs noted weakness in the original submission which seems to have been fixed after the rebuttal period -- primary related to missing experimental details. There was also some concern (as is common with inferential papers) that the claims are difficult to evaluate on real data, as the ground truth is unknown. To this end, the authors provide empirical results with simulated data that address this issue. There is also some concern that more complex predictive models are not evaluated.

Overall the reviewers and AC have a positive opinion of this paper and recommend acceptance.